# Economic shocks predict increases in child wasting prevalence

Derek D. Headey [1✉] & Marie T. Ruel [1]

In low and middle income countries macroeconomic volatility is common, and severe negative economic shocks can substantially increase poverty and food insecurity. Less well understood are the implications of these contractions for child acute malnutrition (wasting), a major risk factor for under-5 mortality. This study explores the nutritional impacts of economic growth shocks over 1990–2018 by linking wasting outcomes collected for 1.256 million children from 52 countries to lagged annual changes in economic growth. Estimates suggest that a 10% annual decline in national income increases moderate/severe wasting prevalence by 14.4–17.8%. An exploration of possible mechanisms suggests negative economic shocks may increase risks of inadequate dietary diversity among children. Applying these results to the latest economic growth estimates for 2020 suggests that COVID-19 could put an additional 9.4 million preschoolers at risk of wasting, net of the effects of preventative policy actions.

[1] The International Food Policy Research Institute (IFPRI), Washington, DC, USA. ✉email: d.headey@cgiar.org

Macroeconomic volatility is far more common in low and middle income countries (LMICs) for a variety of economic, political, and environmental reasons. However, the economic impacts of the COVID-19 pandemic have been exceptional, forcing governments throughout the world to shut down large parts of their domestic and international economies in both 2020 and 2021, thereby transforming a health crisis into a complex economic crisis of exceptional scale, scope, and duration. Virtually all LMIC economies contracted sharply in 2020, and most have experienced further shocks due to the Delta variant in 2021[1]. Other estimates suggest 140 million people may have fallen into $1.90/day poverty in 2020[2].

While there is not yet any evidence to suggest that COVID-19 has directly resulted in significantly higher mortality risks for young children in LMICs, the indirect effects of COVID-related economic shocks pose serious risks for malnutrition, morbidity, and mortality, particularly among children under 5 years of age. In the short term, nutritional insults often manifest in the form of acute weight loss, typically measured as low weight-for-height z scores (WHZ) that are used to classify wasting into different levels of severity. Wasting is usually the result of both severe reductions in food intake and recent or repeated episodes of infectious diseases, and the dynamic interaction between poor diets and infections[3]. Infants and young children are at the greatest risk of wasting – and of mortality due to wasting–because of their immature immune system and their high nutrient requirements for growth and development. Although less prevalent than stunting–an indicator of chronic undernutrition–wasting is a much stronger predictor of child mortality. A pooled analysis of ten prospective cohort studies estimated that severe wasting had a hazard ratio of 11.6 compared to 5.5 for severe stunting (height-for-age Z-scores < −3), while moderately wasted children (−2 > WHZ > −3) were 3.4 times and mildly wasted children (−1 > WHZ > −3) 1.6 times as likely to die before their fifth birthday compared to non-wasted children[4].

Despite the serious risks that wasting poses for morbidity and mortality in young children, the underlying economic causes of wasting are under-researched compared to the many studies linking longer-term economic growth to stunting[5–11], or the numerous studies estimating the impacts of negative economic shocks on child mortality.[9] Moreover, while LMICs have made significant progress in the past few decades in reducing stunting, progress in reducing wasting is uneven at best, with some programmatic improvements in treating wasting[12], but limited improvement in prevention. Globally, more than 45 million under-5 children were estimated to be moderately (WHZ < −2) or severely (WHZ < −3) wasted in 2020, and as much as 75% of wasted children residing in South Asia and sub-Saharan Africa (particularly the Sahel and Horn of Africa), where prevalence of wasting remains remarkably high[13].

In this study, we explore the impacts of short-term economic growth shocks on the risks of child wasting. While are data are historical and pre-date COVID-19, understanding whether the economic shocks currently being experienced in LMICs are likely to significantly increase risks of wasting is an important motivation of this study. To do so we use an extensive set of 177 Demographic Health Surveys (DHS)[14] that collected information on 1.256 million children in 52 countries over three decades (1990–2018). We link these child and household level data to national level estimates of short- and medium-term changes in Gross Domestic Product per capita (GDP) or Gross National Income per capita (GNI), which is equal to GDP plus net income earned abroad. With some modifications, we follow the empirical strategies of previous papers[15,16] that combined macroeconomic data with multiple rounds of DHS data to examine the mortality effects of economic shocks: fixed effects estimators that control for region-specific time trends to net out the biasing effects of many potential confounding factors.

We show that departures of annual economic growth from long term growth trends (our technical definition of economic shocks) increase the risk of child wasting in the subsequent calendar year, and by a significant margin: the model predicts that a 10% decline in GNI per capita predicts a 14.4–17.8% increase in moderate/severe wasting in children under-5. Our findings also support the hypothesis that increases in child wasting are associated with increased risks of children having poor dietary diversity and infectious diseases symptoms such as diarrhea or fever. Moreover, applying these growth elasticities to the latest GDP growth estimates for 2020[1] predicts that an extra 9.4 million children could become wasted in 2021 as a result of COVID-related economic contractions in 2020.

Although this estimate may be an upper bound given the various interventions that governments have implemented to protect their economies during the pandemic, the effectiveness of these interventions in preventing household economic hardship and wasting remains uncertain. Unless protected urgently and holistically, children in LMICs could well be neglected victims of an economic pandemic affecting poor populations throughout LMICs, with no immediate end in sight.

## Results

**Outline**. Our results are structured in three parts: (1) Descriptive results designed to outline the data structure and patterns in wasting and economic shocks; (2) Main regression results for child wasting; (3) predicted impacts of COVID-19 on wasting; (4) sensitivity tests and extensions; and (5) an exploration of potential mechanisms.

**Descriptive results**. Our sample of 1.256 million children 0-59 months of age spread across 52 LMICs exhibits large variation in wasting prevalence (See Supplementary Tables 1–3 for more sample details). Figure 1 reports moderate/severe wasting prevalence for children 0–35 m for the most recent DHS round in all surveyed countries according to established national thresholds of low/very low (<5%), medium (5–10%), high (10-15%) and very high (>15%) wasting prevalence.[17] Wasting is very common in South Asia (notably India) as well as the Sahel and parts of the Horn of Africa, but generally much lower in the rest of sub-Saharan Africa despite many African countries having high rates of stunting, a measure of chronic malnutrition. Supplementary Fig. 1 reports moderate/severe wasting prevalence by child age for each region in sub-Saharan Africa and Asia using local polynomial estimates that allow for flexible wasting-age dynamics. We find strikingly distinct differences in the progression of wasting between Asian and Africa children. In South Asia and South-East Asia peak wasting is highly prevalent at birth (around 27%), consistent with previous evidence on low birth weight and poor maternal weight gain during pregnancy in South Asia,[18] (27%), but then declines steadily to level off at around 15% from age 3 onwards. In sub-Saharan Africa wasting progression follows a very different pattern: children are commonly born wasted, wasting rates are then typically steady for the first few months of life, but then rise and peak in the 10-12 months of age (at 11–17% in most sub-regions) before declining thereafter. However, in the Sahel these dynamics are noticeably accentuated: 21% of children are born wasted, but by 11 months wasting prevalence reaches 30% and thereafter declines to 10% by 36 months.

The 177 DHS rounds in our dataset cover a wide range of recent economic growth shocks. We use both GDP and GNI shocks because the two measures do not perfectly agree nor

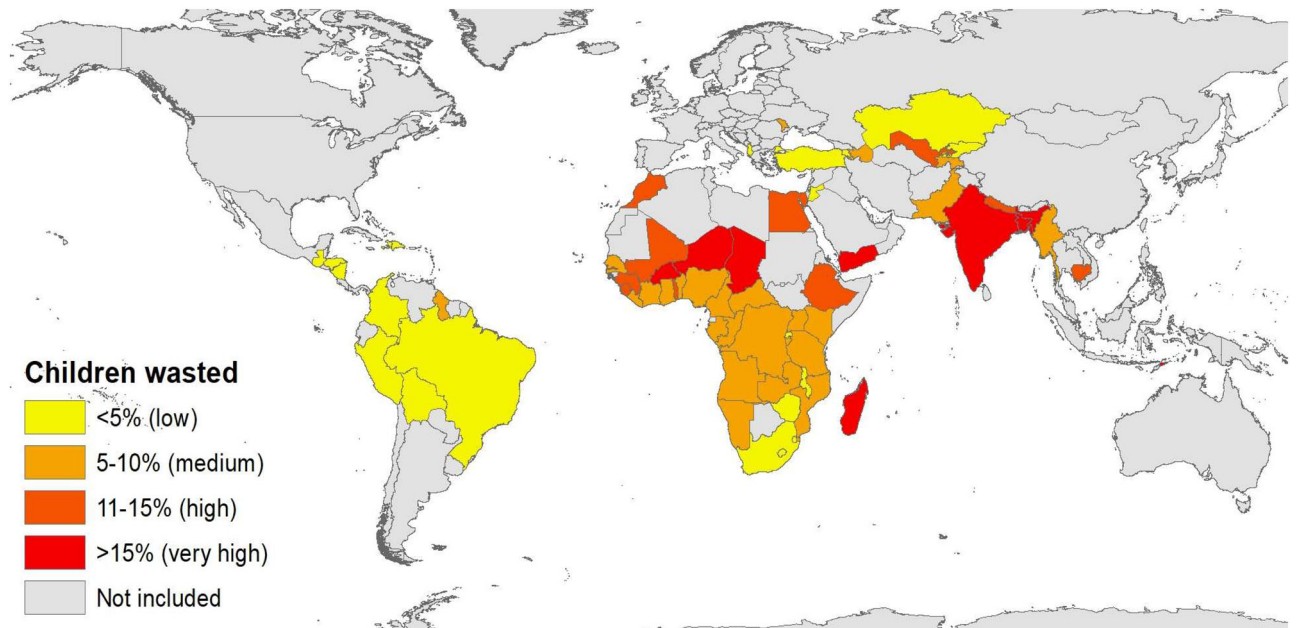

**Children wasted**

- <5% (low)
- 5-10% (medium)
- 11-15% (high)
- >15% (very high)
- Not included

**Fig. 1 Prevalence of moderate/severe wasting among children 0–35 months of age.** Authors' construction from survey-weighted estimates for children 0–35 m of age in 67 countries. Data pertain to the most recent DHS round in each country.

**Table 1 Weighted multivariate linear probability models of wasting risks as a function of lagged annual change in GNI per capita or GDP per capita (children 0–59 months).**

|  | Any wasting (WHZ < −1) | Moderate/severe wasting (WHZ < −2) | Severe wasting (WHZ < −3) |
|---|---|---|---|
| Elasticity of GNI shocks ($w.g^n$)[a] | (1) | (2) | (3) |
|  | −0.071*** | −0.144*** | −0.222*** |
|  | (−0.114, −0.028) | (−0.213, −0.076) | (−0.325, −0.118) |
| R-squared | 0.119 | 0.068 | 0.033 |
| Elasticity of GDP shocks ($w.g^n$)[b] | (4) | (5) | (6) |
|  | −0.088*** | −0.178*** | −0.291** |
|  | (−0.162, −0.013) | (−0.312, −0.044) | (−0.526, −0.056) |
| R-squared | 0.119 | 0.068 | 0.033 |
| DHS child, mother, household effects?[c] | Yes | Yes | Yes |
| Country fixed effects? | Yes | Yes | Yes |
| Region-specific temporal effects?[d] | Yes | Yes | Yes |

$N = 1,256,076$ children in all regressions. Results are linear probability model coefficient point estimates with 95% confidence intervals based on standard errors clustered at the country level reported in parentheses, with significance levels as follows: ***$p < 0.01$, **$p < 0.05$, *$p < 0.10$. Regressions are weighted to be representative of the <5 year population of children of all countries included in this DHS dataset through a three-step weighting procedure factoring in country <5 year population size, DHS round sample size, and conventional DHS survey weights. The coefficients in these regressions can also be interpreted as elasticities as follows: [a]This elasticity is the coefficient on the interaction between annual change in GNI per capita lagged one year and the country-specific prevalence of wasting averaged over all DHS rounds, defined for each specific wasting indicator. The coefficient, therefore, represents the elasticity of wasting. [b]This is analogous coefficient/elasticity for lagged growth in GDP per capita. The regressions control for various factors not reported for the sake of brevity: [c]DHS child, maternal, and household effects include household asset ownership, maternal education years, piped water and flush toilet access, whether the child was born in a medical facility, whether the mother received four or more antenatal care visits, whether the child received all vaccinations, whether the child was born of a teenage pregnancy, whether the mother has four or more children, whether the child is female, and resides in a rural area. [d]Temporal effects include region-specific seasonality effects using month of survey dummies, wasting-age dynamics captured by child age dummies, and time trend effects captured by 5-year time dummies.

produce identical statistical results, and because for some countries net income earned abroad can constitute a significant addition to gross domestic product. Supplementary Fig. 2 and Supplementary Fig. 3 report histograms showing the frequency distribution of lagged annual changes in GNI and GDP for the 177 DHS surveys. The lagged average annual growth rate was positive for both GNI and GDP but there was large variation in growth rates across country-year dyads, and roughly one quarter of these DHS surveys were preceded by negative GNI or GDP growth in the preceding year, with negative shocks especially common in sub-Saharan Africa. Household-level data reported in Supplementary Table 1 also show how vulnerable this sample of children is to adverse economic shocks: 34% of the sample owned none of five listed assets and just 7% owned all five assets; education, health, and demographic indicators are consistent with very low levels of development; diarrhea and fever were also

common in the two weeks prior to the interview, 12% of mothers had low BMI, and less than one-third of children 6-35 m achieved minimum dietary diversity (a proxy for dietary quality).

**Main regression results**. Our main multivariate linear probability model results for child wasting are reported in Table 1, which shows results for any wasting (WHZ < −1), moderate/severe wasting (WHZ < −2), and severe wasting (WHZ < −3) for the aggregate sample of 1.256 million children 0-59 months of age. The model follows Eq. (1) in the *Data and methods* by controlling for a wide range of child, maternal and household level covariates from the DHS (such as assets, education, water, sanitation, and demographics), country fixed effects, and various region-specific temporal effects to control for wasting-age dynamics (as in Supplementary Fig. 1), seasonality in wasting, and secular wasting

trends. The main explanatory variable of interest, however, is the interaction between the lagged annual change in GNI or GDP per capita and mean wasting prevalence in a country, the coefficients of which we term growth elasticities. Standard errors are clustered at the country level, and Table 1 reports the derived 95% confidence intervals.

The principal finding is that GNI and GDP growth elasticities for all forms of wasting are negative and highly statistically significant. In absolute magnitude, the growth elasticity increases with the severity of wasting, although there are sizable differences in point estimates across GNI and GDP growth. For GNI, a 10 percentage point reduction from a country's long term annual growth rate predicts a 7% increase in mild wasting, a 14% increase in moderate/severe wasting, and 22% increase in severe wasting, while the corresponding impacts for GDP shocks are 8.8%, 17.8%, and 29.1%. As we show below, these implied effects are large, suggesting that wasting is highly responsive to adverse growth shocks, although we note that the confidence intervals for these elasticities are reasonably large, especially for GDP shocks.

**Predicted impacts of COVID-19 on wasting**. We next demonstrate the magnitude of these elasticities by using them to predict the impacts of the economic growth shocks experienced in 2020 on wasting in 2021 (to reflect the 1-year lag in the statistical model). Results from an earlier version of this study were also used to estimate the potential global change in numbers of wasted children using September 2020 GNI growth projections from IFPRI[19]. Here we use the latest GDP growth estimates for 2020 from the April 2021 IMF Outlook[1] (Table 2), but also report disaggregated results by region. Since our econometric model measures growth shocks as deviations in annual economic growth from a long term average, we use the IMF Outlook data to measure growth shocks as the difference between GDP growth estimates for 2020 and each country's average country-specific GDP growth rate over 2010-2019. This shock is then combined with the GDP shock elasticity of −0.178 for moderate/severe wasting reported in regression (5) in Table 1.

In total, the model estimates that an addition 9.37 million preschool children could have become wasted in 2021 as a result of the growth shocks experienced in 2020. It is also notable that despite the use of a slightly different growth elasticity (GDP instead of GNI), a different measure of growth shocks (GDP deviations from long-run trends instead of a counterfactual GNI loss), and a slightly different sample size, our results are very close to the 9.3 million projection previously reported using GNI shocks in combination with IFPRI MIRAGRODEP GNI projections for 2020[19].

One commonality across projections is the huge impact of large negative economic growth in India, which suffered an unusually large GDP shock in 2020 (−14.9%), had exceptionally high under-5 wasting prevalence prior to COVID-19 (20.8%), and an exceptionally large population of children under the age of 5 (almost 117 million). Clearly, the massive risk of increased wasting in India is an issue that warrants closer empirical investigation and extraordinary policy responses, especially in light of the extremely severe third wave of COVID-19 cases in the second quarter of 2021.

A final point of some note is that these estimates likely represent upper bounds, as they reflect historically derived wasting-growth elasticities. The COVID-19 crisis motivated a number of extraordinary social protection measures designed to bolster household incomes, employment, and small businesses, and these may have protected nutritionally vulnerable households and their children to some extent.

**Table 2 Estimating the potential increase in moderate/severe wasting for 104 LMICs in 2021 based on the magnitude of GDP growth shocks in 2020.**

| | N (countries) | Growth shock, 2020[a] | Wasting prevalence, 2019[b] | Predicted wasting prevalence, 2021[c] | Wasted children in 2019 | Predicted wasted children, 2021 | Change in number of wasted children |
|---|---|---|---|---|---|---|---|
| Europe & Central Asia | 11 | −7.2% | 3.6% | 4.0% | 595,327 | 649,894 | 54,567 |
| Latin America & Caribbean | 22 | −9.3% | 2.8% | 3.2% | 1,454,980 | 1,669,074 | 214,094 |
| Middle East & N. Africa | 12 | −9.1% | 7.1% | 8.0% | 3,166,284 | 3,364,147 | 197,863 |
| East Asia, excluding China | 10 | −9.2% | 7.8% | 9.1% | 5,497,099 | 6,324,098 | 826,999 |
| Sub Saharan Africa | 43 | −6.5% | 7.7% | 8.4% | 13,113,190 | 14,320,514 | 1,207,324 |
| South Asia, excluding India | 5 | −6.0% | 12.4% | 13.7% | 5,504,399 | 5,910,794 | 406,394 |
| India | 1 | −14.9% | 20.8% | 26.3% | 24,297,263 | 30,759,634 | 6,462,371 |
| Total | 104 | | | | 53,628,543 | 62,998,155 | 9,369,612 |

Source: N = 104 countries. Authors' predictions are based on simulations derived using the regression results in Table 1, the size of the growth shocks reported in Table 1, in conjunction with the GDP growth elasticity of −0.178 for moderate/severe wasting reported in regression (5) in Table 1. [a]IMF GDP shocks are defined as the difference between GDP growth estimates for 2020 from the April 2021 IMF Outlook[1] and IMF GDP growth estimates averaged over 2010–2019. Definitions and sources are as follows: [a]IMF GDP shocks are measure growth shocks for each country and the pre-COVID wasting prevalence in 2019. [b]Wasting prevalence in 2019 refers to the most recent WHO[45] estimate of moderate/severe wasting prevalence in each country, while the number of wasted children in 2019 is the product of the estimated prevalence in 2019 and the population aged 0–4 years in 2020 from the UN population database[35]. [c]Predicted wasting is the product of 2019 wasting prevalence and the growth elasticity for GDP form regression (5) in Table 1.

**Table 3 Exploring disease, maternal nutrition and diet mechanisms linking GNI or GDP growth shocks to child wasting.**

| Dependent variable | Diarrhea in past 2 weeks | Fever-only in past 2 weeks | Low maternal BMI | Minimum diet diversity |
|---|---|---|---|---|
| Age range | 0–59 m | 0–59 m | 15–49 years | 6–35 m |
| Elasticity of GNI shocks ($w.g^n$) | (1) | (2) | (3) | (4) |
| | −0.073 | −0.071 | −0.087 | 0.194** |
| | (−0.210, 0.063) | (−0.267, 0.125) | (−0.230, 0.057) | (0.004, 0.382) |
| R-squared | 0.063 | 0.065 | 0.164 | 0.156 |
| Elasticity of GDP shocks ($w.g^n$) | (5) | (6) | (7) | (8) |
| | −0.123* | −0.184* | −0.144 | 0.155 |
| | (−0.274, 0.021) | (−0.387, 0.019) | (−0.368, 0.080) | (−0.091, 0.399) |
| R-squared | 0.063 | 0.065 | 0.164 | 0.156 |
| DHS child, mother, household effects?[a] | Yes | Yes | Yes | Yes |
| Country fixed effects? | Yes | Yes | Yes | Yes |
| Region-specific temporal effects?[b] | Yes | Yes | Yes | Yes |

$N = 1,230,393$ for the diarrhea and fever-only regressions, while $N = 884,436$ for the low maternal BMI regression and $N = 323,014$ for the Minimum diet diversity regressions. Results are linear probability model coefficient point estimates with 95% confidence intervals based on standard errors clustered at the country level reported in parentheses, with significance levels as follows: ***$p < 0.01$, **$p < 0.05$, *$p < 0.10$. All regressions control for country fixed effects as well as region-specific seasonality effects, wasting-age dynamics, and trend effects. Note that these regressions refer to contemporaneous GNI or GDP growth rates rather than lagged growth rates. Regressions are weighted to be representative of the <5 year population of children of all countries included in this DHS dataset. The regressions control for various factors not reported for the sake of brevity: [a]DHS child, maternal and household effects include household asset ownership, maternal education years, piped water and flush toilet access, whether the child was born in a medical facility, whether the mother received four or more antenatal care visits, whether the child received all vaccinations, whether the child was born of a teenage pregnancy, whether the mother has four or more children, whether the child is female, and resides in a rural area. [b]Temporal effects include region-specific seasonality effects using month of survey dummies, wasting-age dynamics captured by child age dummies, and time trend effects captured by 5-year time dummies.

**Robustness tests and extensions**. Additional results testing robustness to variations in specifications and samples are reported in the Supplementary Information. Supplementary Fig. 3 and Supplementary Fig. 4 report sensitivity of the estimates to different specifications, starting with a very basic specification with fixed and temporal effects model but without any other controls, followed by sequentially adding different sets of DHS-based controls, then the addition of national level lagged changes in rainfall, temperature and battle deaths, removal of unusually high or low GNI growth rates, and replacement of the UN-based GNI or GDP growth measure with measures from the World Bank. The estimated elasticities are statistically significant and negative in all regressions and the magnitude of effects remain largely unchanged. One exception is that the World Bank-based GNI growth measure produces elasticities that are substantially larger in absolute magnitude than the UN measure we principally use. Results are also highly robust to subsets of child age (results available on request).

Supplementary Tables 4 and 5 report tests for differential impacts across urban and rural children and boys and girls. There is some evidence that the wasting status of urban children is more sensitive to growth shocks, consistent with the common finding that macroeconomic shocks often have more adverse impacts on the urban poor[20–22], while rural populations (where farming is a principal livelihood) tend to be more protected because of the inelastic demand for food. For GDP shocks, but not GNI shocks, we also find some evidence that the wasting status of girls is somewhat more sensitive to shocks than for boys. This heightened vulnerability of girls to economic shocks is consistent with an earlier literature suggest that girls in India are more adversely affected during times of deprivation[15,23].

**Exploring potential nutrition mechanisms**. Table 3 explores potential mechanisms linking growth shocks to contemporaneous risks of wasting. Both poor diets and infections can lead to wasting (and undernutrition can lead to disease also), and poor maternal nutrition could also affect neonatal weight or have more complex indirect effects, and might proxy for general household food and nutrition insecurity. Supplementary Table 6 reports associations between moderate/severe wasting and child morbidity symptoms and low maternal BMI in models that exclude growth shocks. Child diarrhea and fever prevalence and low

maternal BMI are associated with increased risks of wasting, while the child achieving minimum dietary diversity is associated with a reduced risk of wasting. Hence these are all theoretically plausible mechanisms through which growth shocks could affect child wasting.

The results in Table 3 test whether shocks predict these intermediate outcomes, but the results show substantial sensitivity to the choice of GNI or GDP as the measure of macroeconomic shocks. For GNI shocks, only child minimum diet diversity is statistically associated with growth shocks (at the 5% level); a 10% negative shock predicts that a child is 19.4 points less likely to achieve an adequately diverse diet. For GDP shocks, the minimum dietary diversity elasticity is reasonably large in magnitude but imprecisely estimated (perhaps because of the much smaller sample size as dietary diversity is only measured in more recent DHS surveys) and not statistically different from zero. However, Supplementary Fig. A5 also looks at how a child's consumption of specific food groups is associated with growth shocks. Consumption probabilities of most nutrient-dense foods shows some tendency to decrease with negative growth shocks, although only consumption of flesh foods is significantly associated with growth shocks.

In addition to that suggestive evidence on dietary impacts, GDP shocks are also significantly and negatively associated with diarrhea and fever prevalence, although only at the 10% level, and the result does not robustly extend to GNI shocks. Elasticities of low maternal BMI with respect to growth shocks were imprecisely estimated and not statistically different from zero. Hence, evidence from this analysis on the hypothesis that economic shocks affect child infections or maternal weight is inconclusive.

## Discussion

This study examined the impact of macroeconomic shocks–defined as annual differences from long-run economic growth–on the risk of child wasting in LMICs. We find robust evidence that the elasticity of wasting with respect to lagged growth shocks is negative and relatively large in magnitude. The results are robust to variations in specification, although point estimates for GDP shocks tend to be larger than those of GNI shocks (though less precisely estimated). We find some evidence of significantly greater sensitivity of urban children to macroeconomic shocks, and some evidence that girls are more sensitive to GDP shocks. We also find some

indications that poor diets and increased infectious disease risks might be important mechanisms linking growth shocks to wasting, bearing in mind that the results are sensitive to the choice of GNI or GDP shock measures.

This study makes a novel contribution to the existing literature on economic growth and nutrition, which has been largely confined to studying the impacts of longer term economic growth on child stunting[5–11]. Moreover, since wasting is a major risk factor in early childhood mortality[24], our results help explain previous findings demonstrating that adverse economic shocks in LMICs significantly increase child mortality rates[9].

Our findings are also highly relevant in the context of the complex and ongoing COVID-19 crisis because they predict that aggregate income shocks alone–even without COVID-related health disruptions–will have severe impacts on wasting and increase the risks of elevated child mortality rates[4]. The true extent of these economic shocks has become increasingly apparent, with incorporation of April 2021 IMF GDP growth estimates predicting that approximately 9.37 million children could become wasted in 2021, but also some further deterioration in economic output in LMICs as the Delta variant has spread far and wide. Globally, the impacts of COVID-19 on wasting very much depend on what happens in India (and some other large countries). Disconcertingly, previous research suggests that state-level macroeconomic shocks in India do indeed predict higher levels of mortality among young children, especially girls[15].

Our empirical results also suggest that macroeconomic shocks might raise wasting risks among urban children more than they do among rural children, consistent with recent evidence on the impacts of COVID-19[25] as well as welfare evidence from earlier macroeconomic crises in LMICs populations[20–22]. This is likely because the farm economy is relatively resilient to macro shocks, with demand for food being less adversely affected than demand for most non-farm goods and services.

This study has limitations. Although we have a large sample of children, the number of economic shocks we study (country-year dyads) is relatively small and offers limited opportunities for exploring heterogeneity across shocks or across countries. Macroeconomic shocks can be caused by very diverse underlying factors (we at least try to net out the potentially confounding effects of poor weather and conflict), and the effects of any given shock could affect the welfare of different socioeconomic groups quite heterogeneously, depending on factors such as economic structure and the extent of social protection measures. Of course, if the impacts of economic shocks are indeed highly heterogenous, this might lead to attenuation of the relevant regression coefficient, perhaps towards statistical insignificance. The fact that we do find such statistically significant coefficients in the wasting regressions gives us some confidence that acute malnutrition is indeed generally quite sensitive to recent macroeconomic shocks in a general sense, especially where wasting is prevalent in normal times.

Another limitation is that it is also well known that national income measures are imperfect predictors of household welfare[26], and indeed some early country case studies of COVID-19's economic impacts suggest that losses in household income will be much larger than losses in national income averaged over the population[27]. Contextually, the COVID-19 crisis has also generated unprecedently large government responses to limit the economic damage to households, although the scale of the response in LMICs has thus far been inadequate relative to the economic damage[28].

A final limitation is that this study only assessed the impacts on wasting due to economic shocks. While economic channels are clearly important in 2020 and 2021, the prolonged COVID-19 crisis will also affect child wasting and mortality through health system disruptions, including suspension of a range of essential nutrition and health actions such as antenatal care, vitamin A

supplementation, and immunization, and the prevention and treatment of severe acute malnutrition and infections[19,29,30]. Reassignment of health staff to COVID-19 tasks, restrictions on mobility, lack of transport, and fear of using health services during lock-downs will also affect access to and utilization of health and nutrition services. A recent COVID-19 study estimated that different scenarios of reduced coverage of health and nutrition services combined with assumptions about increases in wasting could result in 168,000 additional deaths in under-five children in LMICs (or 283,000 in a more pessimistic recovery scenario)[19]. That study estimated that increased wasting would account for close to 25% of the additional deaths.

Our study complements these findings by identifying an important channel of impact: negative economic shocks substantially increasing the risk of all types of wasting. This implies that efforts to protect health systems also need to be bolstered by nutrition-sensitive social protection programs to protect incomes of poor and vulnerable populations, including newly poor populations who will often reside in urban areas. Another potential problem is that wasting is often seen as a public health issue rather than a broader problem of underdevelopment (unlike stunting), although our results suggest that wasting is very much affected by economic shocks, not just health-related problems. Wasting and other important forms of malnutrition are multi-dimensional problems requiring concerted multisectoral solutions, especially in the wake of COVID-19. Unfortunately, COVID-19 continued to ravage many LMICs in 2021 and early 2022 (the time of writing). Much stronger international support is needed to solve the immediate health and nutrition crises affecting these countries, as well as the ongoing economic crises affecting LMICs.

## Methods

**Data**. To explore the impact of macroeconomic shocks on child wasting we combined a large multi-country child-level DHS dataset with national-level macroeconomic data. The DHS are well suited for this kind of analysis because of their high degree of standardization and coverage of a wide range of LMICs, their collection of a rich array of nutrition, health, demographic and socioeconomic data, their representativeness at both national and subnational levels, and their repeated application within countries over different periods of time. Our dataset comprises 177 DHS rounds that collected data on child weight for children 0–59 months of age in 52 LMICs between 1990 and 2018 (See Supplementary Table 1). We note that DHS has excellent coverage of sub-Saharan Africa and South Asia, the two regions with the highest rates of wasting, but is under-represented in South-East Asia where wasting rates are also relatively high (e.g. Indonesia, the Philippines). Even so, the surveys are representative of approximately 400 million under-5 children.

DHS data were used to calculate WHZ scores relative to WHO reference weight-for-height measures of healthy breastfed children in multiple countries[31]. We then defined three standard measures of wasting: any wasting (WHZ < −1), moderate/severe wasting (WHZ < −2), and severe wasting (WHZ < −3).

Our key explanatory variable is the lagged annual change in economic growth, although economic growth can be measured through both GNI and GDP. GDP measures the value of goods and services produced within a country and includes national output, expenditures, and income, while GNI equals GDP plus wages, salaries, and property income of the country's residents earned abroad and at home. A priori, it is not obvious which is a better predictor of changes in household incomes, and the correlation between the two indicators in our sample, while high ($r = 0.90$), is imperfect. In this study we sourced both of these indicators from the UN National Accounts Database,[32] although we also tested robustness to a World Bank source for these same indicators[33].

The remaining variables in our analysis are control variables specified to minimize the bias of confounding factors or to explore non-income channels linking COVID-19 to increased wasting risk. To use a comparable measure of wealth across a wide range of countries we developed a simple classification of ownership of five assets: improved flooring, electricity, TV, fridge and car/motorbike. We classified households into three levels of ownership: no assets, some assets, all five assets. We controlled for maternal education, three proxies designed to capture the continuum of maternal and child health care (antenatal care (% mothers who attended ≥4 visits in previous pregnancy)), medical facility births and vaccinations (% children fully immunized for age), improved sanitation and water supplies, household demographics (teenage births, high fertility rates ≥4 children), child sex and rural location. In addition to these DHS controls we also followed a

previous cross-country DHS study on growth shocks by testing sensitivity to two additional national-level controls that could influence economic growth but independently affect child health: climate shocks, as captured by the one-year lag of total annual rainfall and average monthly temperature[34]; and conflict shocks, captured by battle-related deaths per 100,000 people[33].

Finally, we used child morbidity symptoms and dietary indicators, as well as low maternal body mass index (BMI < 18.5) to explore whether disease, dietary or maternal nutrition mechanisms might explain the impacts of growth shocks on child wasting. For morbidity symptoms we use the DHS measure of whether a child was reported to have had diarrhea or fever-only in the past two weeks. For diets we use the standard minimum dietary diversity (MDD) indicator corresponding to a child having consumed at least four of seven food groups in the previous 24 h measured for children 6–35 months. We also use the individual food groups to test sensitivity of their consumption to growth shocks.

**Methods**. Our analysis of these data was conducted in three stages. All analysis was conducted in STATA™ Version 16.

First, we estimated a new set of household weights to make the regressions more representative of under-5 children in the DHS countries in our analysis. In a typical single-country analysis one need only use the standard DHS survey weights for each household to render statistics nationally representative, but in a multi-country analysis a more complex approach is needed, especially when the main explanatory variable of interest is itself measured at the country-year level. This is because: (a) countries differ in terms of their total population of under-5 children; (b) different countries have different numbers of rounds, and survey sizes that do not reflect population sizes; and (c) different rounds within a country can have very different sample sizes. Were we not to construct explicit weights to reflect these facts then the regression coefficients for growth shocks would remain implicitly weighted by the somewhat arbitrary number of observations in each individual survey and the number of surveys in each country, resulting in some growth episodes being over-weighted or under-weighted relative to their country's population of under-5 children or relative to the number of survey rounds per country.

To address this, we constructed a three-step weighting metric. First, we used United Nations[35] statistics on the population of children <5 years of age to create a country-level population weight. India, for example, accounts for 20% of all child-level observations in our sample, but India contains 34% of all under-5 children in our DHS sample of countries (Supplementary Table 2). Second, we re-weighted observations within rounds to correct for imbalances in sample sizes. To continue the Indian example, its 2015-16 DHS round has 232,761 child observations while its 2005-06 round has just 42,615 (Supplementary Table 1), so the new weight corrects this imbalance to apply equal weight to both rounds. Finally, we used the standard DHS survey weights to ensure representativeness within surveys.

In the second step of our analysis we used different descriptive analysis techniques to explore patterns and trends in the data by mapping moderate/severe wasting, using non-parametric regressions (the *lpolyci* command in STATA) to plot wasting by child age and region, and examined the distribution of growth shocks across DHS rounds.

Third, we used a multivariate linear probability model to test the impacts of lagged growth shocks conditional upon long-run wasting prevalence. By interacting lagged GNI or GDP shocks with the average wasting prevalence across surveys we allow the effect of economic shocks to be linearly proportional a country long-run wasting prevalence. This is biologically appropriate (as populations in which wasting is more prevalent should be more vulnerable to negative shocks), but also mathematically appropriate since a WHZ sample that is distributed more closely to the various wasting thresholds should see larger absolute changes in wasting. This specification also has a benefit for interpretation since the coefficient represents the elasticity of wasting prevalence with respect to economic growth shocks. The one-year lag in growth shocks is also justified on several grounds. First, the timing of wasting measurement varies within years but growth shocks can occur at different times within a year; specifying a lagged growth shock always ensures that the macroeconomic shock precedes wasting measurements. Second, caregivers may take actions to prevent or delay the translation of economic shocks into nutritional insults for their children, so the impacts of economic shocks may be quite delayed. Third, although diets and diseases could emerge quickly in the wake of economic shocks, changes in weight can a longer period of time.

The remaining variables in the model are control variables. The regression model closely approximates a difference-in-difference model in that it uses fixed and temporal effects to net out numerous time-invariant factors that influence wasting as well as common trends over time, such as improvements and expansions in programs aimed at preventing and treated acute malnutrition. This approach is similar to a previous study examining the impacts of growth shocks on retrospective mortality measures from the DHS[16], although the application to wasting poses some additional challenges highlighted below.

The first is that wasting is seasonal and seasonal variations in wasting are region-specific. In South Asia, the available evidence suggests that wasting sharply increases in the monsoon[36–38], whereas in some parts of the Sahel and Horn of Africa it is reported to increase in drier seasons[39,40]. Second, and relatedly, wasting may have quite different etiologies in different regions. For example, low birth weight is more prevalent in South Asia than in sub-Saharan Africa despite similar poverty levels.[41] Moreover, DHS evidence suggests that there may be difference in

long-term wasting trends across regions; wasting rates have also been stubbornly immune to longer-term economic growth in South Asia, but have declined notably in some West African countries, such as Ghana.

To control for these complex differences in the trends and timing of wasting, we first created a more refined series of regional dummy variables that account for the diversity of wasting prevalence within sub-Saharan Africa: the Sahel and Horn of Africa (where wasting is highly prevalent), Western and Central Africa (where wasting is moderately prevalent), and Eastern and Southern Africa (where wasting is rare, despite high rates of stunting and general poverty). The remaining regions are more standard: South Asia, South-East Asia, Middle East, North Africa, Eastern Europe, Central Asia, Latin America, and the Caribbean. We then interacted these regional dummy variables with three kinds of temporal effects: 5-year time brackets to flexibly control for secular changes in wasting in different regions; month of child measurement dummies to control for seasonality of wasting; and child age dummies (in months) to control for differences in the region-specific progression of wasting in early life.

With the control variables outlined above, the linear probability models take the form:

$$w_{i,c,r,t} = \beta_0 + \beta_g \bar{w}_{c,r} g_{c,r,t-1} + \beta_X X_{i,c,r,t} + \beta_C C + \beta_A A \cdot R + \beta_S S \cdot R + \beta_T T \cdot R + \varepsilon_{i,c,r,t}$$

(1)

This equation posits that wasting ($w$) for child i in country c and region r at time t is a function of lagged annual economic growth ($g_{t-1}$) interacted with average wasting prevalence across all rounds ($\bar{w}$). We note that mean wasting ($\bar{w}$) refers to means of each specific wasting indicator (mild, moderate, severe), depending on which is specified on the left-hand side of Eq. (1), and also that $g$ is re-scaled to the unit of a 10 percent change in GNI or GDP to facilitate interpretation of the coefficients as elasticities.

The remaining variables in Eq. (1) include a vector of control variables from the DHS ($X$), country fixed effects ($C$), and three types of region-specific temporal effects: child age effects ($A.R$), seasonality effects ($S.R$), and trend effects ($T.R$). Since the specification includes country fixed effects, the coefficient on economic growth represents the elasticity of wasting risks with respect to deviations of lagged growth from long-term growth rates, which we, therefore, define as growth shocks. Standard errors ($\varepsilon$) are clustered at the country level for the calculation of 95% confidence intervals (CIs) in all tables, although several graphs report 90% CIs to reflect one-sided tests of the null hypothesis that the growth elasticities are significantly below zero.

Although all our regressions follow the basic structure of Eq. (1) we conduct several sample restrictions and specification modifications.

First, we explored the influence on key results of excluding various sets of control variables, as well to exclusion of extreme growth outliers. Our assumption is that many DHS controls are pre-determined and not influenced by recent growth shocks, including household assets, parental education, sanitation, and water sources, but access to health services might indeed be seriously affected by shocks.

Second, we tested variation in wasting impacts by child age since previous research has shown that wasting prevalence is higher in young children (e.g. <2 years) and that wasting risks in younger children appear are more sensitive to various nutritional insults[42]. However, little variation across age ranges was uncovered, so those results are not reported herein (results are available on request).

Third, we introduced interactions with urban and girl dummies to test for differential impacts of growth shocks on boys and girls and rural and urban children since previous research on the health and nutrition impacts of economic shocks suggests impacts may differ by gender[15,23,42,43] and that urban populations are typically more vulnerable to economic crises than rural populations[20–22].

Finally, we explored potential mechanisms linking growth shocks to wasting, including symptoms of common childhood infections such as diarrhea and fevers, maternal nutrition (low BMI), as well as minimum dietary diversity for children 6–35 months of age, and its individual food group components. These regressions follow the same structure as Eq. (1), but with disease or dietary diversity as dependent variables, while contemporaneous growth interacted with country means for these new dependent variables rather than with mean wasting levels. We also estimated wasting regressions with disease symptoms, low maternal BMI, and dietary diversity as explanatory variables to further test the validity of these mechanisms.

**Reporting summary**. Further information on research design is available in the Nature Research Reporting Summary linked to this article.

## Data availability
The data that support the findings of this study are available from The Demographic Health Surveys (https://dhsprogram.com/data/available-datasets.cfm) but restrictions apply to the availability of these data, which were used under license for the current study, and so are not publicly available. Data are however available from the authors upon reasonable request and with permission of the DHS. UN and World Bank macroeconomic indicators are publicly available. The authors are committed to sharing their data with any interested researchers.

## Code availability
The STATA™ code for replication of the results is available on ZENODO.[44]

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

## Acknowledgements

The authors thank various members of the "Standing Together for Nutrition" group for comments and suggestions, as well as David Laborde, Will Masters, Kalle Hirvonen, Harold Alderman, Giordano Palloni and Dennis Petrie for sharing their ideas. We also thank Wahid Quabili and Zhe Guo for research assistance. All errors are our own. DH is grateful to The Bill and Melinda Gates Foundation for financial support through the *Advancing Research on Nutrition and Agriculture (ARENA) project, Phase II (Investment ID: OPP1177007)* and both authors received support from the CGIAR program on Agriculture for Nutrition and Health (A4NH).

## Author contributions

DH and MR conceptualized the study and wrote the article. DH conducted the empirical analysis, with inputs from MR.

## Competing interests

The authors declare no competing interests.
