## [Peer Review File · Nature Communications]

Economic shocks predict increases in child wasting prevalenceREVIEWER COMMENTS

Reviewer #1 (Remarks to the Author):

Referee Report – The impact of economic recessions on child acute malnutrition: Implications for the COVID-19 Crisis
Nature Communications: NCOMMS-20-22867

This is a very timely paper that addresses a fundamentally important issue. The results are of potentially broad interest. The author's bottom line is that economic recessions in general cause large increases in child wasting, and thus the current COVID-19 pandemic portends substantial increases in wasting. While this core intuition is compelling, I have a number of questions about the econometric analysis through which the authors arrive at that conclusion. For simplicity, I'll list those questions in the order they occur as I read through the text.

The analysis treats economic shocks as homogeneous phenomena. This seems problematic in a paper that combines data from 52 countries over a period of nearly 40 years – especially if the goal is to connect recessions to child wasting. My concern here relates to the question of mechanisms (more below on this). Intuitively, one would expect the effect of a recession due to, say, a famine to have quite a different connection to and impact on wasting than a recession due to inept monetary policy or a drop in oil prices for an oil exporting country. What is it about the COVID-19 pandemic in particular – and the specific nature of the resulting economic shock in LMICs that connect it to wasting? A homogeneous representation of economic recessions seems a very blunt "treatment" variable when the specific mechanisms of transmission from macro to micro could lead to important differences in wasting outcomes.

Is the result specific to the use of GNI versus other national income aggregates? I ask because the paper specifically – and incorrectly – distinguishes GNI from GDP by saying that the former differs from the latter by excluding public and private investment. It's correct that GDP includes investment; however, $GNI = GDP + \text{net factor income from abroad}$ -- and thus still includes investment. (I'm not saying GNI is the wrong choice, but the given justification vaguely suggests the results weren't robust to using GDP. There's no national income aggregate that excludes investment.)

Can the author provide some assurance about the parallel trends assumption that's required for identification in difference-in-difference models? Has the author tested this assumption?

With regard to the other control variables – I like the inclusion of rainfall and temperature; however, I wonder whether it wouldn't make more sense to enter them as anomalies relative to trends. If a country is always hot/dry, would that necessarily explain wasting in panel data?

The paper uses the terms GNI, growth, recession, and shocks more or less interchangeably. This leads to some confusion about the specific hypothesis being tested (e.g., recession has a specific definition that doesn't fit with lagged GNI). The author should more clearly distinguish year-to-year changes in GNI from growth shocks, and clarify what is being tested. Since the author frames the empirical analysis as a difference-in-difference model, is the treatment variable a growth shock? If so, define it. (The discussion refers to the "small number of economic shocks" – it's not clear how this is modelled, or whether that small number refers both to positive and negative shocks.)

By restricting the analysis to wasting as a binary outcome, is there information being lost that might be apparent using WHZ as a continuous outcome variable?

The author describes an innovative weighting scheme for the data. Are the results sensitive to this re-weighting?

The core estimating equation (1) is confusing in its presentation. g^n and g^p are not clearly defined. Is the former a negative change in lagged GNI and the latter positive? Does this suggest

potentially different effects depending on the direction of change? (If so, why?) It's not clear what is meant by the rescaling of these variables as 10% changes in GNI. Are they just divided by 10, so that a 1-unit change represents a 10% change? (If so, why?) In a fixed effects model, can these be interpreted as first differences in lagged GNI, while wasting is a level effect? And, if the specification is regressing a level dependent variable against changes in the treatment variable, is that formally a difference-in-difference model?

The author should explain why g^n is interacted with mean wasting. It suggests an unstated hypothesis that the effect of negative economic growth is greater when there is generally a higher prevalence of wasting. Why might that be the case? And why not similarly interact positive changes in GNI? Given only this interaction term, it's also difficult to interpret the result – is it driven by GNI or by w ? (Also, if the mean prevalence of wasting is constant within countries, what does this interaction term mean in a fixed effects regression?)

The estimates are clustered at the DHS cluster level. However, the observations are sampled with respect to the treatment variable at the country level. Should clustering be at the country level, since observations are not independent within countries with respect to the treatment?

This issue of independence within countries is illustrated in Table 1. The author has 1.2 million observations of wasting, but only 177 observations on the national-level indicators (not 1.2 million as indicated in Table 1).

The results presented in Table 2 are very interesting. However, it's confusing to see the results for g^n but not for g^p (which relates to point 8 above). It's not clear what the author specifically means by a growth elasticity if it follows from changes in only one direction. (And here, too the distinction between "growth" and "growth shocks" may be relevant.) Also, should one be concerned about endogeneity between wasting and explanatory variables such as whether a child was born in a medical facility, taken for ANC visits, and vaccinated?

Regressions with 1.2 million observations tend to find everything statistically significant. Often, the suggestion in such cases is to focus more on effect size. The paper could be more detailed in assessing the effect size proposed.

The attention to mechanisms of transmission from growth shocks to wasting is an important and interesting aspect of the paper. But a few questions still arise.

Table 4 is described as estimating the effect of growth shocks on diarrhea and other child morbidities related to wasting, but in the table the treatment variable is described as the growth elasticity – previously defined in Table 2 as a function of the interaction of GNI and mean wasting. Does that mean that regressions 1-4 in Table 4 include wasting (as distinct from a pure economic growth shock) as part of the explanation for diarrhea? That's confusing – especially as regressions 5 and 6 model wasting as a function of diarrhea, etc.

The author indicates in a footnote that GNI in Table 4 is modeled as contemporaneous in its effect on morbidities, whereas in the wasting regressions GNI was lagged. Why the change in Table 4? The earlier explanation for using lagged GNI was households could cope with income shocks in the short run, suggesting potential lags between the shock and impacts on wasting. However, the modeling now posits that income shocks have a lagged effect on wasting, but a contemporaneous effect on the mechanisms that connect the shock to wasting. In essence, the goal is to estimate the connection between an income shock in time $t-1$ to wasting in time t . The regressions in Table 4 suggest that: a) an income shock in time $t-1$ causes morbidity in time $t-1$, and b) morbidity in time t causes wasting in time t . But this chain of estimates is broken by the need for morbidity in $t-1$ to cause morbidity in time t , which isn't established (and isn't obviously plausible).

In this context, it would add to the strength of the analysis to have more detail in explaining how a shock to national income leads to heightened diarrhea, etc. (Here, some differentiation in the source of the income shock could be relevant.)

The motivation for the paper is that the COVID-19 pandemic could impact child wasting. The

connection of the pandemic per se to wasting as described in the paper is mostly implied, and depicted as an income shock. Is there anything more specific to the pandemic's effect, even at the national level, that could connect to wasting? Would it be relevant to dive more deeply into the specific macroeconomic effects of the pandemic (e.g., it's both a supply and a demand shock) in thinking about its potential impacts on child wasting? Are there specific effects on food supply or demand in affected countries?

Reviewer #2 (Remarks to the Author):

This is a well-conceived analysis of a large multi-country child-level DHS dataset in relation to lagged annual changes in Gross National Income. The analysis appears robust and the results are of importance to the international community in responding to the COVID-19 crisis. I have relatively few comments.

Both the prevalence and risks associated with wasting are dependent on socio-economic status and, in-turn vary nationally. The authors acknowledge that the impact may vary by socio-economic status and give examples of when the risk of wasting is substantially reduced. However, I would like to have seen a more substantial overall result for how disproportionately more and less advantaged children may be affected. For example, illustrating the elasticity of wasting with GNIpc growth across levels of assets and WASH (as is illustrated for age in the appendix). This would be critically important information for targeting responses within countries and provide evidence to support the discussion comment '...to protect incomes of poor and vulnerable populations'.

Under limitations, are there other factors that may be confounders that have changed over time, such as changes in ANC care, roll out of community-based screening and management of malnutrition, and in some areas, HIV? A potential bias is the lack of survey data in unstable or conflict regions (Sudan, Somalia etc). To what extent do the DHS surveys include displaced people?

From a policy and finance perspective, an important problem is the traditional siloed approach of stunting being regarded as a chronic issue to be dealt with by development agencies whilst wasting is seen as a public health issue. In the discussion there is an opportunity to provide a data-led comment on this issue.

Dear Editor,

Both reviewers make a number of important suggestions for revising the paper. We have responded to the reviewers point by point in blue italics below and pasted the relevant text where necessary. Here we note a few major changes suggested by Reviewer 1:

1. We now reported country-clustered standard errors throughout.
2. We now report results using both GNI and GDP to measure macro shocks, as the results are at least quantitatively affected by this.
3. We now interpret magnitude of the coefficients/elasticities, by using them to project the increased number of wasted children that our statistical model predicts will result from the COVID-19 macroeconomic shocks.

There are a number of other smaller changes we address below and in some instances we just provide a response to the reviewers' queries rather than adding more results to the document itself. Overall, though, the reviewers' comments were very constructive and their suggestions have undoubtedly improved the manuscript.

Warm regards

The authors

REVIEWER COMMENTS

Reviewer #1 (Remarks to the Author):

This is a very timely paper that addresses a fundamentally important issue. The results are of potentially broad interest. The author's bottom line is that economic recessions in general cause large increases in child wasting, and thus the current COVID-19 pandemic portends substantial increases in wasting. While this core intuition is compelling, I have a number of questions about the econometric analysis through which the authors arrive at that conclusion. For simplicity, I'll list those questions in the order they occur as I read through the text.

The analysis treats economic shocks as homogeneous phenomena. This seems problematic in a paper that combines data from 52 countries over a period of nearly 40 years – especially if the goal is to connect recessions to child wasting. My concern here relates to the question of mechanisms (more below on this). Intuitively, one would expect the effect of a recession due to, say, a famine to have quite a different connection to and impact on wasting than a recession due to inept monetary policy or a drop in oil prices for an oil exporting country. What is it about the COVID-19 pandemic in particular – and the specific nature of the resulting economic shock in LMICs that connect it to wasting? A homogeneous representation of economic recessions seems a very blunt “treatment” variable when the specific mechanisms of transmission from macro to micro could lead to important differences in wasting outcomes.

This is an excellent point that we have thought about quite a bit, though unfortunately we think there is a limited amount we can empirically do about it. There are certainly many types of economic

contractions, and even within broad types (e.g. macroeconomic shocks) there are undoubtedly variations that could result in different impacts depending on the exact nature and timing of the shock, the structure of the economy and the livelihoods of its vulnerable population and the protection policies in place. We are clearly only estimating an average effect across heterogeneous impacts. We also agree that the COVID19 crisis could indeed be different from previous shocks, although for the most part it has been a massive demand-side shock due to losses of employment and income, and we have seen these before in Africa (e.g. CFA Franc devaluation in 1994) and Asia during the Asian financial crisis (1997-98). However, we do note that we did test sensitivity of our results to climate shocks and conflict shocks, because these could obviously create lower economic growth, but would constitute qualitatively different economic crises (e.g. cause public service disruptions).

A further point to note is that if the effects of growth shocks on wasting were wildly heterogeneous then the coefficients would be very imprecisely estimated and perhaps even attenuated. So, not being able to differentiate impacts by type of shocks is certainly a limitation but we can still have some confidence that growth shocks generally lead to child wasting. We have made this point in the manuscript.

We think the best way to address your comment is just to acknowledge the nature of the estimate: it is a cross-country average that could cover up heterogeneous impacts, and it may be an upperbound estimate if countries have reasonable social protection and SAM treatment measures in place during COVID-19. We have modified the text accordingly. The discussion section now includes the following text:

“This study admittedly has limitations. Although we have a large sample of children, the number of economic shocks we study (country-year dyads) is relatively small and offers limited opportunities for exploring heterogeneity across shocks or across countries. Macroeconomic shocks can be caused by very diverse underlying factors (we at least try to net out the potentially confounding effects of poor weather and conflict), and the effects of any given shock could affect the welfare of different socioeconomic groups quite heterogeneously, depending on factors such as economic structure and the extent of social protection measures. Of course, if the impacts of economic shocks are indeed highly heterogeneous, this might lead to attenuation of the relevant regression coefficient, perhaps towards statistical insignificance. The fact that we do find such statistically significant coefficients in the wasting regressions gives us some confidence that acute malnutrition is indeed generally quite sensitive to recent macroeconomic shocks in a general sense, especially where wasting is prevalent in normal times.”

Is the result specific to the use of GNI versus other national income aggregates? I ask because the paper specifically – and incorrectly – distinguishes GNI from GDP by saying that the former differs from the latter by excluding public and private investment. It’s correct that GDP includes investment; however, $GNI = GDP + \text{net factor income from abroad}$ -- and thus still includes investment. (I’m not saying GNI is the wrong choice, but the given justification vaguely suggests the results weren’t robust to using GDP. There’s no national income aggregate that excludes investment.)

Apologies for this error. Our reasoning for GNI was to include the effects of income earned abroad (to some extent remittance income is captured in GDP, but of course, many countries have large populations earning income overseas). But the question of robustness to GDP is valid, and it’s also useful to add GDP to the analyses since the IMF forecasts/estimates use GDP in its Outlook reports. We find that the results

are robust to GDP instead of GNI, though that is not surprising – the two series of GNI and GDP shocks are very highly correlated (0.90). We now report both GDP and GNI shocks in the paper – for all results – and use 2020 GDP growth estimates from April 2021 IMF Outlook to estimate changes in wasting in 2021.

	(1) N=1,256,076 Any wasting (WHZ < -1)	(2) N=1,256,076 Moderate/severe wasting (WHZ < -2)	(3) N=1,256,076 Severe wasting (WHZ < -3)
Growth elasticity ($w.g^n$), with GNI	-0.071*** (-0.092, -0.050)	-0.144*** (-0.185, -0.103)	-0.222*** (-0.293, -0.151)
	Any wasting (WHZ < -1)	Moderate/severe wasting (WHZ < -2)	Severe wasting (WHZ < -3)
Growth elasticity ($w.gn$), with GDP	-0.088*** (-0.162, -0.013)	-0.178*** (0.312, -0.043)	-0.291*** (-0.526, -0.056)

Can the author provide some assurance about the parallel trends assumption that's required for identification in difference-in-difference models? Has the author tested this assumption?

Unfortunately we don't see a way to implement a conventional parallel trends test because of a number of somewhat unusual features of the model and data. The model is not a panel on the left hand side (they are different cross-sectional samples of children), and the model focuses on short term (annual) shocks, not secular growth income. The data are also highly unbalanced with different surveys in different years and some countries having relatively few DHS rounds (i.e. 2-3). These concerns made us realize that we should not call the estimator a diff-in-diff, but rather emphasize it approximates some of the features of a diff-in-diff by netting out both fixed effects and many temporal confounders.

With regard to the other control variables – I like the inclusion of rainfall and temperature; however, I wonder whether it wouldn't make more sense to enter them as anomalies relative to trends. If a country is always hot/dry, would that necessarily explain wasting in panel data?

Because the model controls for country fixed effects, the log country-level rainfall and temperature in the year before survey interview effectively is the deviation between rainfall/temperature in the year prior to the survey relative to the long term (fixed) rainfall/temperature. However, we experimented with other ways of measuring rainfall and temperature and the results on GNI and GDP were unaffected.

The paper uses the terms GNI, growth, recession, and shocks more or less interchangeably. This leads to some confusion about the specific hypothesis being tested (e.g., recession has a specific definition that doesn't fit with lagged GNI). The author should more clearly distinguish year-to-year changes in GNI from growth shocks, and clarify what is being tested. Since the author frames the empirical analysis as a difference-in-difference model, is the treatment variable a growth shock? If so, define it. (The discussion refers to the "small number of economic shocks" – it's not clear how this is modelled, or whether that small number refers both to positive and negative shocks.)

We apologize for the confusion and agree that we have not been precise enough in our first version of the paper. In effect the model specifies the effect of the difference between last year's growth rate and long-run annual growth rates (captured by fixed effects). Hence the coefficient refers to the departure of recent economic growth from the long-run growth rate, which we can think of as a series of shocks of different magnitudes; some trivial, some large. Of course, we are interested in growth episodes that could be regarded as shocks in common parlance: i.e. unusually large positive or negative growth episodes. In terms of C19 and major economic contractions more generally, it's clearly the big negative shocks that we are interested in. We have kept the title focused on economic shocks, but now defined very clearly what the right hand side is measuring, as described above. We have dropped references to recessions. Thank you for this point – it will improve the clarity of the paper.

By restricting the analysis to wasting as a binary outcome, is there information being lost that might be apparent using WHZ as a continuous outcome variable?

Yes, in principle using binary outcomes could be problematic, although in this instance we have a lot of data for the left-hand side variable, so lack of precision is unlikely to be problematic. Moreover, the main reason for using dichotomous variables is that mild, moderate and severe wasting are associated with different risks of child mortality; so using changes in whz across the whole distribution would be misleading and would underestimate risk for children with lower WHZ. A relevant publication on this (already cited in the paper) is:

Olofin, I., McDonald, C.M., Ezzati, M., Flaxman, S., Black, R.E., Fawzi, W.W., Caulfield, L.E., Danaei, G., 2013. Associations of suboptimal growth with all-cause and cause-specific mortality in children under five years: a pooled analysis of ten prospective studies. PLoS One 8, e64636.

For completeness, we also estimated elasticities for WHZ as the dependent variable. As expected, this elasticity is negative and statistically significant. For GNI shocks the elasticity is -0.089 (CI -0.177, -0.001) and is significant at the 5 level, while for GDP shocks it is significant at the 2% level -0.106 (CI -0.193, -0.018). However, for the reasons above we do not report these results in the main text, but we hope we have satisfactorily addressed the reviewer's concern.

The author describes an innovative weighting scheme for the data. Are the results sensitive to this re-weighting?

*Qualitatively, the results are not sensitive as the signs are all the same and the elasticities are still statistically significant, but the magnitudes are much smaller (about half as small in the case of GNI). However, as noted in the text, the "unweighted" results are still weighted implicitly, but the weights are arbitrarily based on DHS samples rather than population representativeness. Hence, we prefer not to present these results, even in the appendix, as they are quite misleading. For any, moderate/severe and severe wasting the GNI elasticities are -0.035**, -0.078** and -0.145**, while for GDP they are -0.055***, -0.103** and -0.15*.*

The core estimating equation (1) is confusing in its presentation. $g^{\Delta n}$ and $g^{\Delta p}$ are not clearly defined. Is the former a negative change in lagged GNI and the latter positive? Does this suggest potentially different effects depending on the direction of change? (If so, why?) It's not clear what is meant by the rescaling of these variables as 10% changes in GNI. Are they just divided by 10, so that a 1-unit change represents a 10% change? (If so, why?) In a fixed effects model, can these be interpreted as first differences in lagged GNI, while wasting is a level effect? And, if the specification is regressing a level dependent variable against changes in the treatment variable, is that formally a difference-in-difference model?

We apologize for several typos in the equation. We have made edits to the equation and text below to explain the equation better. With fixed effects, the lagged annual growth in GNI or GDP represents the elasticity of deviations of recent economic growth from long-run economic growth, which we define as economic shocks (of varying size). The temporal effects also control for trends, seasonality, and wasting-age dynamics. As noted above, we agree that this cannot be called a formal difference-in-difference model but is perhaps close to an approximation of such models. We mention this more explicitly, and have removed all references to the model as a diff-in-diff.

The re-scaling of GNI or GDP changes to make the coefficient more interpretable as an elasticity.

The author should explain why $g^{\Delta n}$ is interacted with mean wasting. It suggests an unstated hypothesis that the effect of negative economic growth is greater when there is generally a higher prevalence of wasting. Why might that be the case? And why not similarly interact positive changes in GNI? Given only this interaction term, it's also difficult to interpret the result – is it driven by GNI or by \bar{w} ? (Also, if the mean prevalence of wasting is constant within countries, what does this interaction term mean in a fixed effects regression?)

The hypothesis was stated in the text:

“By interacting lagged GNI shocks with the average wasting prevalence across surveys we allow the effect of changes in GNI to be linearly proportional to a country long-run wasting prevalence. This is biologically appropriate (as populations in which wasting is more prevalent are likely to be more vulnerable to negative shocks) but also mathematically appropriate since a WHZ distribution that is distributed more closely to the various wasting thresholds should see larger absolute changes in wasting. This specification also has a benefit for interpretation since the coefficient represents the elasticity of wasting prevalence with respect to economic growth.”

We find the interaction works well because in early versions when we were doing sample restrictions by mean wasting prevalence, we found that most of the effects were driven by countries with larger increases in wasting (>5% prevalence). That approach was cumbersome because it relied on a somewhat arbitrary threshold. The interaction with mean wasting produced better fits of the data. With country fixed effects there is no problem introducing an interaction between growth shocks and mean wasting, although mean wasting itself is absorbed by the country fixed effects.

The estimates are clustered at the DHS cluster level. However, the observations are sampled with

respect to the treatment variable at the country level. Should clustering be at the country level, since observations are not independent within countries with respect to the treatment?

Yes, this is a great point and we agree. We have now reported standard errors based on country level clustering through the text. This does not affect the main results, but some of the results on disease and diets are affected and no longer significant at the 10% levels (though coefficients are unchanged obviously).

This issue of independence within countries is illustrated in Table 1. The author has 1.2 million observations of wasting, but only 177 observations on the national-level indicators (not 1.2 million as indicated in Table 1).

Yes, good point; we have now clarified this is in the relevant table (now appendix table A1).

The results presented in Table 2 are very interesting. However, it's confusing to see the results for $g^{\wedge n}$ but not for $g^{\wedge p}$ (which relates to point 8 above). It's not clear what the author specifically means by a growth elasticity if it follows from changes in only one direction. (And here, too the distinction between "growth" and "growth shocks" may be relevant.) Also, should one be concerned about endogeneity between wasting and explanatory variables such as whether a child was born in a medical facility, taken for ANC visits, and vaccinated?

As explained above, this was a typo in equation (1), which assumes that negative and positive growth shocks have proportionately similar effects (our 177 country-year observations are not really powered to test this assumption, unfortunately).

As noted above, we have clarified exactly what is meant by growth shocks in this statistical set-up.

Those endogeneity concerns are valid for analyses of those DHS variables, but obviously our focus is really just on growth shocks, and we already show that the growth shock elasticities are very robust to inclusion/exclusion of control variables in the DHS. In the main text we now only report growth shock elasticities to save space and minimize distraction for the reader.

Regressions with 1.2 million observations tend to find everything statistically significant. Often, the suggestion in such cases is to focus more on effect size. The paper could be more detailed in assessing the effect size proposed.

This is a great point and we agree. We address this point in two ways. First, we report country-clustered standard errors since other approaches might exaggerate the degree of precision. Second, we introduce a new table showing the projected impact of growth shocks in 2020 due to COVID on wasting prevalence and numbers of wasted children in 2021 in order to interpret the size of these effects. The GNI and GDP results predict increased wasting of around 9 million children, most of them in India. We think this adds a lot of value to the paper. Thank you for the suggestion.

The attention to mechanisms of transmission from growth shocks to wasting is an important and

interesting aspect of the paper. But a few questions still arise.

Table 4 is described as estimating the effect of growth shocks on diarrhea and other child morbidities related to wasting, but in the table the treatment variable is described as the growth elasticity – previously defined in Table 2 as a function of the interaction of GNI and mean wasting. Does that mean that regressions 1-4 in Table 4 include wasting (as distinct from a pure economic growth shock) as part of the explanation for diarrhea? That’s confusing – especially as regressions 5 and 6 model wasting as a function of diarrhea, etc.

Apologies for the confusion. The growth in GNI and GDP is not interacted with wasting in these regressions, but with the mean of each dependent variable (e.g. diarrhea, fever, low BMI, MDD). We have clarified this now in the text.

The author indicates in a footnote that GNI in Table 4 is modeled as contemporaneous in its effect on morbidities, whereas in the wasting regressions GNI was lagged. Why the change in Table 4? The earlier explanation for using lagged GNI was households could cope with income shocks in the short run, suggesting potential lags between the shock and impacts on wasting. However, the modeling now posits that income shocks have a lagged effect on wasting, but a contemporaneous effect on the mechanisms that connect the shock to wasting. In essence, the goal is to estimate the connection between an income shock in time t-1 to wasting in time t. The regressions in Table 4 suggest that: a) an income shock in time t-1 causes morbidity in time t-1, and b) morbidity in time t causes wasting in time t. But this chain of estimates is broken by the need for morbidity in t-1 to cause morbidity in time t, which isn’t established (and isn’t obviously plausible).

We understand this concern, but we have consulted with other nutritionists on this, and they agree that there likely is a significant lag between economic shocks and wasting. In contrast, we know from recent COVID-19 empirical analyses that households can sometimes change diets quite quickly in response to income shocks (the poor, especially, as they have few other coping mechanisms sometimes), and we have added references to that effect. But then the translation of poorer diets or more disease into wasting could be a more prolonged process. In the Indonesian financial crisis, for example, several studies found that diets changed quickly - with marked declines in consumption of nutrient dense foods (e.g. eggs) - but child wasting rates increased several points somewhat later on in the crisis. Still, we make note of this uncertainty in the discussion section now.

Block, S., Kiess, L., Webb, P., Kosen, S., Moench-Pfanner, R., Bloem, M.W., Timmer, C.P., 2004. Macro shocks and micro outcomes: child nutrition during Indonesia’s crisis. *Economics and Human Biology* 2, 21-44.

In this context, it would add to the strength of the analysis to have more detail in explaining how a shock to national income leads to heightened diarrhea, etc. (Here, some differentiation in the source of the income shock could be relevant.)

We do feel the theory and evidence behind dietary mechanisms is a bit stronger, but there is also a literature showing that negative growth shocks lead to declining public and household-level health expenditures (Simms and Rowson 2003). In India, Bhalotra (2011) found that economic crises lead

mothers/caregivers to work more, so we may see a deterioration in childcare and hygiene. There are also interactions between diets, immune system and morbidity.

Simms, C., Rowson, M., 2003. Reassessment of health effects of the Indonesian economic crisis: donors versus the data. *The Lancet* 361, 1382-1385.

Bhalotra, S., 2010. Fatal fluctuations? Cyclicity in infant mortality in India. *Journal of Development Economics* 93, 7-19.

The motivation for the paper is that the COVID-19 pandemic could impact child wasting. The connection of the pandemic per se to wasting as described in the paper is mostly implied, and depicted as an income shock. Is there anything more specific to the pandemic's effect, even at the national level, that could connect to wasting? Would it be relevant to dive more deeply into the specific macroeconomic effects of the pandemic (e.g., it's both a supply and a demand shock) in thinking about its potential impacts on child wasting? Are there specific effects on food supply or demand in affected countries?

This is a pertinent question, and as mentioned above we have added new projections of the impact of COVID-19 growth shocks on wasting. There are other studies that outline the mechanisms through which COVID-19 will affect nutrition, including four citations below that we have now added. So far, it seems that demand-side disruptions through income losses is the most important effect, although there are localized supply-side disruptions (mostly somewhat short-term in nature during lockdown periods, particularly in April-May 2020). As we note in the discussion section, there are also serious concerns about disruptions to supply of and demand for health/nutrition services, which could compound the effects of economic shocks.

Akseer, N., Kandru, G., Keats, E.C., Bhutta, Z.A., 2020. COVID-19 pandemic and mitigation strategies: implications for maternal and child health and nutrition. *The American journal of clinical nutrition*.

Roberton, T., Carter, E.D., Chou, V.B., Stegmuller, A.R., Jackson, B.D., Tam, Y., Sawadogo-Lewis, T., Walker, N., 2020. Early estimates of the indirect effects of the COVID-19 pandemic on maternal and child mortality in low-income and middle-income countries: a modelling study. *The Lancet Global Health* 8, e901-e908.

Headey, D., Heidkamp, R., Osendarp, S., Ruel, M., Scott, N., Black, R., Shekar, M., Bouis, H., Flory, A., Haddad, L., Walker, N., 2020. Impacts of COVID-19 on childhood malnutrition and nutrition-related mortality. *The Lancet* 396, 519-521.

Osendarp, S., Akuoku, J. K., Black, R. E., Headey, D., Ruel, M., Scott, N., . . . Heidkamp, R. (2021). The COVID-19 crisis will exacerbate maternal and child undernutrition and child mortality in low- and middle-income countries. *Nature Food*. doi:10.1038/s43016-021-00319-4

Reviewer #2 (Remarks to the Author):

This is a well-conceived analysis of a large multi-country child-level DHS dataset in relation to lagged annual changes in Gross National Income. The analysis appears robust and the results are of importance to the international community in responding to the COVID-19 crisis. I have relatively few comments.

Thank you for the positive feedback.

Both the prevalence and risks associated with wasting are dependent on socio-economic status and, in turn vary nationally. The authors acknowledge that the impact may vary by socio-economic status and give examples of when the risk of wasting is substantially reduced. However, I would like to have seen a more substantial overall result for how disproportionately more and less advantaged children may be affected. For example, illustrating the elasticity of wasting with GNIpc growth across levels of assets and WASH (as is illustrated for age in the appendix). This would be critically important information for targeting responses within countries and provide evidence to support the discussion comment ‘...to protect incomes of poor and vulnerable populations’.

This is an excellent suggestion and we had indeed initially explored this a bit by looking at the split across asset levels in addition to the rural-urban split. However, we tend to see some (relatively weak) evidence that the kids in the poorest households in asset terms are not quite as badly affected as other kids. This may be explained by the fact that in macro situations it is moderately poor urban households who are most affected. The poorest households tend to be rural and engaged in farming, and farming is known to be reasonably robust to macro shocks because demand for food is more resilient than demand for other goods and services, and poor people can typically fall back on farming. The text provides some citations to support this claim from the Indonesian financial crisis and earlier crises in West Africa too, as well as COVID-related economic simulation studies. However, we should add that we don’t have a great deal of precision to tease out heterogeneity in impacts.

On the specific results, the GNI shock elasticity for kids in households that are not asset-poor is -0.208 (CI -.296, -0.120), while for asset-poor households it is -0.1062 (-0.173, -0.038). We get similar coefficients for GDP shocks: for non asset-poor households’ kids the coefficient is -0.281 (CI -0.476, -0.084) while the elasticity for children of asset-poor households is -0.119 (CI -0.214, -0.025). So in both cases children in asset-poor households seem a bit less vulnerable to macro shocks, but the data below seem to suggest why – extreme asset poverty is 45% in rural areas but just 10% in urban areas. Because of this we think it’s more important to emphasize the rural-urban dichotomy since this has clear targeting implications: i.e. there could be lots of newly-poor households in urban areas, and their children will be vulnerable to the effects of these macro shocks. We have added some more explicit text to the discussion section to reflect this.

	Asset-poverty rate
Rural areas	45.5%
Urban areas	10.3%

Under limitations, are there other factors that may be confounders that have changed over time, such as changes in ANC care, roll out of community-based screening and management of malnutrition, and in some areas, HIV? A potential bias is the lack of survey data in unstable or conflict regions (Sudan, Somalia etc). To what extent do the DHS surveys include displaced people?

This is an excellent point, and we have added a sentence in the limitations section to reflect this issue: “Contextually, the COVID-19 crisis has also generated unprecedentedly large government responses to limit

the economic damage to households, although the scale of the response in LMICs has thus far been inadequate relative to the economic damage.³⁰ .”

To our knowledge the DHS never purposively over-samples IDPs, but doesn't avoid them either. However, it is not possible to identify them in the samples. More generally, there may be selection problems if DHS rounds are less likely to be implemented during or after a macro crisis for some reason. We have mentioned this now also.

From a policy and finance perspective, an important problem is the traditional siloed approach of stunting being regarded as a chronic issue to be dealt with by development agencies whilst wasting is seen as a public health issue. In the discussion there is an opportunity to provide a data-led comment on this issue.

This is a great point. In our final paragraph we have added the following sentence:

“Wasting, moreover, is often seen as a public health issue rather than a broader problem of underdevelopment (unlike stunting), but our results suggest that wasting very much stems from economic shocks, not just health-related problems”

REVIEWERS' COMMENTS

Reviewer #1 (Remarks to the Author):

I appreciate the seriousness the authors have taken in responding to my comments and suggestions. I'm mostly convinced by their responses. However, I'd like to push back on their decision, in response to my previous comment, to now include both GDP and GNI as indicators of income shocks. As a minor observation, the response letter on this point makes reference to remittances being included in GNI. Strictly speaking that isn't correct, as national income accounts distinguish net factor income from abroad and net unilateral transfers (which is where remittances fall, and which is the difference between GNI and gross national disposable income). But my primary reaction to this issue is that the authors should simply choose one. The results aren't so different that the distinction adds to the findings and the authors don't really clarify a relevant functional distinction between GDP and GNI.

For two reasons, I'd suggest going with GDP. One reason is that it's not so clear that net factor income from abroad would be so relevant a mechanism for transmitting domestic income shocks to child wasting. The second reason is that this revision includes a very nice new addition in Table 2 of projected increases in wasting numbers by region -- based on IMF indicators that themselves use GDP. Why not be consistent?

Since Table 2 is new, I have one further suggestion. The approach taken here is to use region-specific growth shocks and then to apply a single elasticity parameter to all regions. This assumes that the elasticity itself is the same for all regions. I'm not suggesting that the authors need to estimate region-specific elasticities, but it might be nice to note this assumption somewhere.

Reviewer #2 (Remarks to the Author):

This is an important topic irrespective of COVID-19 and the analysis is well conceived and executed. It is significant in the field and presents new data of policy and scientific importance, especially highlighting wasting as a consequence of underdevelopment.

The changes in response to the first round of reviews, especially use of GNI and GDP, and examining elasticities, have considerably improved the paper and I have no further concerns about the methods or interpretation. My comments are minor and relate to additional data that could help policymakers and agencies to take the 'urgent preventive actions' suggested.

Analysis of flexible wasting-age dynamics is mentioned. Do economic shock affect wasting in different age groups differently? This could influence policy in relation to timing of additional mitigating interventions in pregnancy (birth weight), focusing on exclusive breast feeding, complementary feeding or screening after the typical ages of clinic contact for infant immunizations.

Several interactions, associations and sub-groups were examined, including urban/rural, maternal BMI, minimum dietary diversity and diarrhea/fever. However, key information that may help guide policy is whether effects may have been mediated by changes in household resources such as income or assets that may influence ability spend on higher quality foods or medicines, or mother's own work or income. Assets data exist within DHS datasets and it would be helpful to know if these fluctuate with GNI/GDP, and what is the effect on the wasting coefficients of their inclusion in the models. I.e., does wasting variation occur independently of household wealth indicators? I realize that these were among the excluded control variables so I am questioning that assumption (as the authors do in their response to the first round of reviews in relation to the Indonesian financial crisis).

'That wasting is often seen as a public health issue rather than a broader problem of underdevelopment (unlike stunting)' is an absolutely critical policy point raised by this analysis. This point about the need to integrate approached has been made before and could be referenced, e.g. <https://gh.bmj.com/content/5/11/e003023.long>, as well as strengthening this point.

REVIEWER COMMENTS

REVIEWERS' COMMENTS

Reviewer #1 (Remarks to the Author):

I appreciate the seriousness the authors have taken in responding to my comments and suggestions. I'm mostly convinced by their responses. However, I'd like to push back on their decision, in response to my previous comment, to now include both GDP and GNI as indicators of income shocks. As a minor observation, the response letter on this point makes reference to remittances being included in GNI. Strictly speaking that isn't correct, as national income accounts distinguish net factor income from abroad and net unilateral transfers (which is where remittances fall, and which is the difference between GNI and gross national disposable income). But my primary reaction to this issue is that the authors should simply choose one. The results aren't so different that the distinction adds to the findings and the authors don't really clarify a relevant functional distinction between GDP and GNI.

For two reasons, I'd suggest going with GDP. One reason is that it's not so clear that net factor income from abroad would be so relevant a mechanism for transmitting domestic income shocks to child wasting. The second reason is that this revision includes a very nice new addition in Table 2 of projected increases in wasting numbers by region -- based on IMF indicators that themselves use GDP. Why not be consistent?

On the basis of the suggestion from the editor we have decided to report both sets of results. The point estimates have the same sign but differ in magnitudes, and when exploring the mechanisms there is some sensitivity to choice of GDP or GNI. Specifically, the editor noted "Please note that the two reviewers differ in their advice about including both GDP and GNI in the main analysis. We suggest that you continue with the current presentation (i.e. using both measures in the analyses)."

Since Table 2 is new, I have one further suggestion. The approach taken here is to use region-specific growth shocks and then to apply a single elasticity parameter to all regions. This assumes that the elasticity itself is the same for all regions. I'm not suggesting that the authors need to estimate region-specific elasticities, but it might be nice to note this assumption somewhere.

Thank you for this point. It is indeed not really possible to test for regional differences in the key growth elasticity, although please note we kind of implicitly do this anyway by estimating a coefficient (elasticity) that is dependent upon the long-run wasting rate, which itself differs by region. Even so, we have made note of the fact that shocks can have heterogenous effects on wasting, and that these shocks are themselves quite heterogenous.

Reviewer #2 (Remarks to the Author):

This is an important topic irrespective of COVID-19 and the analysis is well conceived and executed. It is significant in the field and presents new data of policy and scientific importance, especially highlighting wasting as a consequence of underdevelopment.

The changes in response to the first round of reviews, especially use of GNI and GDP, and examining elasticities, have considerably improved the paper and I have no further concerns about the methods or interpretation. My comments are minor and relate to additional data that could help policymakers and agencies to take the 'urgent preventive actions' suggested.

Analysis of flexible wasting-age dynamics is mentioned. Do economic shock affect wasting in different age groups differently? This could influence policy in relation to timing of additional mitigating interventions in pregnancy (birth weight), focusing on exclusive breast feeding, complementary feeding or screening after the typical ages of clinic contact for infant immunizations.

This was examined in earlier drafts and there were perhaps hints of larger effects on younger children (consistent with their higher wasting prevalence) but there was not enough precision to say the effects were significantly larger, so these were dropped. However, we agree with the reviewer that it is quite likely that younger children are more vulnerable, as indicated by Supplementary Figure 1.

Several interactions, associations and sub-groups were examined, including urban/rural, maternal BMI, minimum dietary diversity and diarrhea/fever. However, key information that may help guide policy is whether effects may have been mediated by changes in household resources such as income or assets that may influence ability spend on higher quality foods or medicines, or mother's own work or income. Assets data exist within DHS datasets and it would be helpful to know if these fluctuate with GNI/GDP, and what is the effect on the wasting coefficients of their inclusion in the models. I.e., does wasting variation occur independently of household wealth indicators? I realize that these were among the excluded control variables so I am questioning that assumption (as the authors do in their response to the first round of reviews in relation to the Indonesian financial crisis).

A good point, but one problem with asset indicators is that they don't tend to go down with negative economic shocks, because they consist partly of things like housing characteristics. We did test – in an earlier draft – for impacts that vary by asset levels, but the results were not strong and likely confounded by the fact that urban populations tend to be better off. Our results weakly suggest that urban populations are harder hit by macro crises, even though rural people are poorer. That is almost certainly because food production is reasonably resilient to macro shocks – people need food even if incomes fall.

'That wasting is often seen as a public health issue rather than a broader problem of underdevelopment (unlike stunting)' is an absolutely critical policy point raised by this analysis. This point about the need to integrate approached has been made before and could be referenced, e.g.

<https://gh.bmj.com/content/5/11/e003023.long>, as well as strengthening this point.

This is an interesting reference, but seems to focus more on the use and misuse of anthropometric indicators and related terminology rather than the policy issues at hand, so we have not referenced it here.